# Multisystemic resilience to shocks: a temporal analysis of health, fundamental rights and freedoms, and economic resilience during the first wave of the COVID-19 pandemic in 22 European countries

Mia Clausin,[1] Alicia Rieckhoff,[1] Fabrizio Tediosi [ID],[2] Chantal M Morel [ID],[1] Yuliya Kaspiarovich,[1] Nicolas Levrat,[1,3] Didier Wernli [ID] [1,4]

MC and AR contributed equally.

For numbered affiliations see end of article.

**Correspondence to**
Prof Didier Wernli;
Didier.Wernli@unige.ch

## ABSTRACT

**Objectives** Research on resilience to the COVID-19 pandemic has primarily focused on health system resilience. The purpose of this paper is to: (1) develop a broader understanding of societal resilience to shocks by evaluating resilience in three systems: health, economic and fundamental rights and freedoms and (2) to further operationalise resilience in terms of robustness, resistance and recovery.

**Settings** 22 European countries were selected based on the availability of data in the health, fundamental rights and freedoms, and economic systems during the first wave of the COVID-19 pandemic in early 2020.

**Design** This study uses time series data to assess resilience in health, fundamental rights and freedoms, and economic systems. An overall resilience was estimated, as well as three of its components: robustness, resistance and recovery.

**Results** Six countries exhibited an outlier excess mortality peak compared with the prepandemic period (2015–2019). All countries experienced economic repercussions and implemented diverse measures affecting individual rights and freedoms. Three main groups of countries were identified: (1) high health and high or moderate economic and/or fundamental rights and freedoms resilience, (2) moderate health and fundamental rights and freedoms resilience and (3) low resilience in all three systems.

**Conclusions** The classification of countries into three groups provides valuable insights into the multifaceted nature of multisystemic resilience during the first wave of the COVID-19 pandemic. Our study highlights the importance of considering both health and economic factors when assessing resilience to shocks, as well as the necessity of safeguarding individual rights and freedoms during times of crisis. Such insights can inform policy decisions and aid in the development of targeted strategies to enhance resilience in the face of future challenges.

## INTRODUCTION

The COVID-19 pandemic has tested the resilience of countries to deal with a shock affecting health, economic, social, environmental

## STRENGTHS AND LIMITATIONS OF THIS STUDY

⇒ The study operationalises measurements of resilience of 22 European countries to the first wave of the COVID-19 pandemic in three systems – health, economic, and fundamental rights and freedoms.
⇒ Three resilience components—robustness, resistance and recovery—in addition to an overall measure of resilience were measured, and different measurements methods were tested.
⇒ The primary limitation when assessing resilience lies in country comparability (countries were not exposed to COVID-19 in the same way) and the interpretation of resilience.
⇒ Selected indicators only partially capture resilience, overlooking other factors influencing countries' capacity to absorb and adapt to a shock.
⇒ While measurement of resilience remains in need of further research, the various measurement methods tested showed significant variations in results.

and governance systems worldwide.[1] In an early 21st century characterised by uncertainty, complexity and systemic disruptions, resilience thinking has gained traction in various fields including disaster management, the environment and more recently global health.[2 3] While current research on resilience to COVID-19 primarily focuses on health systems,[4–6] there is a need to assess resilience beyond health systems.[7] Considering resilience as a multiobjective societal problem, the concepts of 'systemic resilience'[8] and 'multisystemic resilience'[9] have emerged to better understand the impact of shocks across highly interconnected systems.

There are currently different definitions of resilience across fields ranging from the capacity of a system to come back to its initial

state to the capacity '*to absorb disturbance and reorganise while undergoing change so as to still retain essentially the same function, structure, identity, and feedbacks*'.[10] Resilience thinking draws attention to both the vulnerability of systems and the capacity to respond. However, with no universally accepted definition across systems, the measurement of resilience remains underdeveloped.[11] To address this gap, this paper starts from the assumption that some indicators of performance of a system can be used as proxy for the expression of its resilience capacities. The second assumption is that the resilience of a system over time can be divided into different components such as robustness, resistance and recovery.[12 13]

While the policy literature has sought to develop measurement of multisystemic resilience,[14] this cross-country study depicts the first wave of the COVID-19 pandemic in 22 European countries measuring health resilience, economic resilience and resilience of fundamental rights and freedoms. This study provides insights about how different systems are affected by the COVID-19 pandemic. Understanding resilience as the ability of societies to maintain their core governance functions while minimising the undesirable societal effects can serve as the foundation for an integrated approach to build societal resilience to future pandemics.[1]

## METHODS
### Public and patient involvement
This study is based on publicly available data about the impacts of the COVID-19 pandemic on health, individual rights and freedoms, and the economy (the detail of the data used is provided in the section below). No patient, nor the public were involved in the design of the study. No ethical approval was sought as per our university requirements. The STROBE (STrengthening the Reporting of OBservational studies in Epidemiology) guidelines were used for organising the method section.[15]

### Study design and setting
This study investigates the temporal dynamics of resilience of three systems, that is, health, individual rights and freedoms, and the economy, during the initial wave of the COVID-19 pandemic in Europe. An extensive review of academic and intergovernmental sources was conducted to identify relevant indicators of resilience. This search encompassed sources such as the World Bank, WHO, the Organisation for Economic Co-operation and Development (OECD) and European Union. Indicators were selected based on two key criteria: availability of weekly or monthly data capturing temporal evolution and relevance to the specific type of resilience being examined (see the measurements section further). Data collection spanned from week 3, before the implementation of any COVID-19 measures, to week 35, when COVID-19 deaths approached their lowest levels across European countries (13 January 2020–31 August 2020). The analysis focuses on 22 European countries chosen

based on the availability of data pertaining to the three selected systems. Countries lacking data for any of the three systems at the time of data collection were excluded from the analysis. The collected data for each indicator was systematically compiled in a separate spreadsheet for each country.

### Data sources and selected indicators
The indicator selected for measuring health system resilience was weekly excess mortality, from the Human Mortality Database.[16] A death is recorded by week of occurrence, apart from the UK, where death by week of registration is used. Excess mortality was selected as it helps overcome several issues related to the reporting of COVID-19 related deaths such as miscounting from misdiagnosis or under-reporting. It also includes 'collateral damage' from other health conditions left untreated if the health system is overwhelmed.[6] Excess mortality was defined as the number of deaths registered in excess of the 5-year average from 2015 to 2019. The calculations for Germany and Greece were based on the 4-year average (2016–2019) due to unavailable data for 2015. A weekly P-score, which was chosen as it allows for robust country comparisons, was calculated as the percentage difference between the reported and projected number of deaths. Health system resilience was interpreted as the inverse of excess mortality. The lower the excess mortality, the larger health system resilience.

The indicator selected for fundamental rights and freedoms' resilience was the Oxford Coronavirus Government Response Tracker's (OxCGRT) stringency index.[17] The stringency index records the strictness of 'lockdown style' policies that restrict people's behaviour. The index is a composite measure of categorical variables based on nine indicators: school closing, workplace closing, cancelling of public events, restrictions on the size of gatherings, closing of public transport, stay-at-home requirements, restrictions on internal movement, restrictions on international travel and public information campaigns. It therefore provides a proxy indicator to measure restrictions in fundamental rights and freedoms, considering that the latter (eg, freedom of movement; right to assembly and demonstration) are intrinsic to the functioning of liberal democracies. While the indicator is a daily value between 1 and 100 (100=strictest), a weekly average was derived for this study. Resilience of fundamental rights and freedoms was measured as the inverse of the stringency index. This means that the more stringent was the response, the less resilient were fundamental rights and freedoms.

To measure economic resilience, the OECD's short-term main economic indicator: gross domestic product (GDP) ratio to trend was used.[18] It is a monthly derived indicator where GDP, which is a monetary measure of the market value of all goods and services produced and sold in a specific period, is divided by the long-term GDP trend to give a ratio-to-trend. It is notably useful to visualise economic cycles. If the GDP ratio to trend=100, GDP is equal to long-term GDP trends. If the GDP ratio to

**Table 1** Definitions of resilience to COVID-19 and its three core components in health, fundamental rights and freedoms, and the economic systems (inspired by Grafton *et al*[12])

| Definitions | Health system | Fundamental rights and freedoms system | Economic system |
|---|---|---|---|
| Resilience is the capacity of a system to absorb, adapt to and recover from a disturbance | Capacity to meet increases in demand for both public health and healthcare services and adapt to long-standing changes | Capacity to preserve and protect individual rights and freedoms and recover quickly after a shock | Capacity to limit the magnitude of economic losses, recover quickly and forge new developmental paths for prosperity |
| *Robustness* is the capacity of a system to maintain its identity/ performance and not cross an undesirable threshold following an adverse event. | Capacity to maintain health services and population health within the range of normal variation | Capacity to maintain respect for individual rights and freedoms within the range of normal variation | Capacity to maintain economic indicators performance within the range of normal variation |
| *Resistance* is the capacity to absorb disruption with minimal damage to system functionality | Capacity to slow down COVID-19 transmission and excess mortality | Capacity to minimise an escalation of stringent measures violating fundamental rights and freedoms | Capacity to minimise economic disruption and impact of the shock |
| *Recovery* is the capacity to 'bounce-back' following disruption | Capacity to recover to a scenario of prepandemic excess mortality rates | Capacity to end emergency measures that impact fundamental rights and freedoms | Capacity to recover to a scenario with prepandemic GDP trends and avoid a long-term recession |

GDP, gross domestic product.

trend is higher than 100, GDP is higher than long-term GDP trends, which suggests an economic expansion. If the GDP ratio to trend is below 100, the GDP is lower than the long-term GDP trend signalling that the 'growth' (if any) is less than trend. GDP ratio to trend data were compared with quarterly GDP, that is, the percentage change from the same quarter in previous years, to check for its accuracy in representing the magnitude of the shock. Economic resilience to the shock was interpreted as the inverse of the decline of the GDP ratio.

## Measurements of resilience, robustness, resistance and recovery

Table 1 outlines the definitions used for resilience and its three components across the different systems, and figure 1 provides a visual representation of these concepts. Table 2 outlines the criteria used to measure health, fundamental rights and freedoms, and economic resilience, robustness, resistance and recovery. Overall resilience in health, fundamental rights and freedoms

and economic systems was estimated using the cumulative value for the selected measurements. A highly positive value (for excess mortality and stringency index) or negative value (for GDP ratio to trend) means that a country does not express resilience. By contrast, a negative excess mortality value means that the country is exceeding normal times performance (i.e., a resilience overshoot). Results were then used to conduct a comparative analysis across systems and identify general trends and potential trade-offs between them.

Robustness primarily means that a system continues to work at the same level of performance despite a disturbance. For example, the definition of health robustness is the capacity to maintain the same level of health services in case of increased demand.[6] Finding a precise threshold of disruption is challenging as it depends on the system affected and may involve arbitrary considerations. A suitable approach to assess robustness is to evaluate whether the system has continued to demonstrate a level of performance that is in line with previous years. In this study, the 1.5 interquartile range (IQR) method was selected to identify outliers regarding the indicators of health and economic systems resilience. Countries that had values above the 1.5 IQR during the first wave of the COVID-19 pandemic were considered as not robust. Regarding fundamental rights and freedoms, the situation is different as the Oxford stringency index was introduced with the pandemic and there is no point of comparison with the period preceding the pandemic.

Resistance and recovery were estimated using the slope of linear regressions leading to the peak (resistance) and

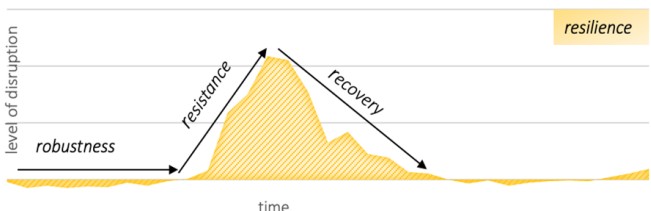

**Figure 1** Conceptual framework of resilience and its components: robustness, resistance and recovery inspired by Grafton *et al*[12].

**Table 2** Measurements and interpretation of resilience, robustness, resistance and recovery across the health – fundamental rights and freedoms, and economic systems

| System | Resilience | Robustness | Resistance rate | Recovery rate |
|---|---|---|---|---|
| Health system | Cumulative excess mortality between 13 January 2020 and 31 August 2020. A low value is indicative of high level of resilience | The value for the period considered is not an outlier for the period 2015–2020 | Slope of linear regression between value closest to 0 and maximum. A low positive value is indicative of high resistance | Slope of linear regression between maximum to first value closest to zero. A high negative value is interpreted as a high level of recovery |
| Fundamental rights and freedoms system | Cumulative stringency index value between 13 January 2020 and 31 August 2020. A low value is indicative of high level of resilience | The value for the period considered is an outlier compared with 2015–2020 | Slope of linear regression between value closest to 0 and maximum. A low positive value is indicative of high resistance | Slope of linear regression between maximum and minimum value. A high negative value is interpreted as a high level of recovery |
| Economic system | Cumulative GDP ratio to trend losses between 13 January 2020 and 31 August 2020. A low value is indicative of high level of resilience | The value for the period considered is an outlier compared with 2015–2020 | Slope of linear regression between value closest to 0 and minimum. A low negative value is indicative of high resistance | Slope of linear regression between minimum and maximum value. A high positive value is interpreted as a high level of recovery |

GDP, gross domestic product.

away from the peak (recovery). We used F-test to assess the statistical significance of the calculated slopes. To select the relevant data for the calculation of a slope, a continuous upward or downward trend was defined as the presence of no more than one subsequent value that does not follow the upward or downward trend. Regarding resistance, we used the interval of data from a minimum (the selected start points was the negative or positive value closest to 0) to the maximum or peak during the period considered from week 3 to 35.[4] For countries that experienced two or more 'peaks', we selected the first peak (as the goal of the paper is to assess resilience at the onset of the pandemic). This means that different countries may have different starting and ending points. A higher slope value (in %) represents a lower level of resistance. For economic resistance, the slope was calculated from the value closest to 100 to the lowest value. Recovery was measured using the slope of the linear regression over the data interval between the maximum value (minimal value for the economic indicator) and the minimal subsequent value. A higher slope value (in %) represents a faster level of recovery.

## Statistical analysis

We conducted an evaluation of association among the measurements of resilience across multiple systems. First, we calculated Pearson correlation coefficients for each system to examine the potential correlation between resistance, recovery and overall resilience. While we successfully computed an overall resilience measure for each country across the three systems, certain combinations of systems and countries did not yield statistically significant results ($p > 0.05$) regarding the correlation between resistance, recovery and overall resilience. Consequently, these countries were excluded from the calculation of associations between resilience and resistance/recovery within the specific system. Second, we calculated Pearson

correlation coefficients to assess whether overall resilience within one system exhibited correlations with resilience in other systems. To facilitate further comparison of resilience across the different systems, we used z-scores to derive two additional metrics. The first metric quantifies the disparity between excess mortality and response stringency, providing an indication of the appropriateness of the response relative to the magnitude of the problem. The second metric measures the deviation from the conventional approach to governance by assessing the difference between an index measuring the state of liberal democracy and the stringency index. The methodology employed for deriving these metrics can be found in online supplemental material 1.

## RESULTS
### Health system resilience

Up to 31 August 2020, six countries exhibited a cumulative excess mortality that was an outlier compared with 2015–2019 (online supplemental material 2 and 3) and were considered as not robust. While the remaining countries were considered as robust during the first wave, three of them experienced peaks above 40% (Netherlands 75%, Switzerland 45%, and Portugal 44%). Several countries reported at least two non-consecutive values above 10% of excess mortality. Among them, only Portugal had two statistically significant peaks. Eight of the nine countries with a mortality peak above 40% also had the highest cumulative excess mortality over the study period, suggesting a low overall health resilience. By contrast, five countries registered negative cumulative excess mortality. These countries were not affected by the first wave of the COVID-19 pandemic to the same extent as countries with high excess mortality. This was possibly due to greater geographical isolation, as in the case of Iceland, or a less important caseload before measures were implemented,

which has been found to be an important factor in the literature.[19]

For the 10 countries for which significant values of both resistance and recovery could be calculated, the peak of excess mortality was reached in 4.8 weeks on average, while the average recovery time was 7.7. Health resistance and recovery were negatively correlated: the lower the health resistance (higher peak), the faster the recovery ($r(10)=0.97$, p<0.001). Spain showed the lowest resistance (45.28%) with excess mortality rates increasing by 6.0-fold in the week of the ninth of March and then almost tripling to 158.25% within 2 weeks. Yet, the country also showed the highest recovery rate (−26.23%). Both low level of resistance ($r(10)=0.86$, p<0.001) and high level of recovery ($r(10)=0.80$, p<0.01) were associated with overall cumulative excess mortality. Spain, with the lowest resistance and highest recovery, also had the lowest expressed health system resilience (653.18%). Yet, low resistance does not necessarily imply low overall resilience. Switzerland exhibited a mortality peak and rather low resistance of 14.5% but with a cumulative excess mortality of only 108.57%, similar to other non-peaking countries like Austria (109.00%) or Greece (119.68%). This can be explained by the country's high recovery rate (−9.60%). With a resistance close to Switzerland's, Sweden (13.8%) had a lower recovery rate (−3.74%) as well as higher cumulative excess mortality (278.98%), compatible with the implementation of less stringent government interventions.

### Resilience of fundamental rights and freedoms
Given the unprecedented level of restrictions implemented in all countries during the first wave of the COVID-19 pandemic in Europe, none can be considered as robust in terms of fundamental rights and freedoms. This is corroborated by further data that showed a further decline in democracy compared with previous years.[20 21] As of 24 August 2020, no country had returned to its prepandemic situation. Italy, Portugal, UK and Spain exhibited the highest cumulative stringency over the study period, while Sweden, Estonia, Iceland and Finland exhibited the lowest cumulative stringency (online supplemental material 4). The average cumulative fundamental rights and freedoms resilience for all 22 countries was 1478.72. Countries often maintained a high stringency level regardless of their epidemiological situation during the first wave, as reflected in the low fundamental rights and freedoms recovery average of −3.20 points compared with health average recovery rate of −9.36% (for countries with a statistically significant value for the calculation of health system recovery).

No relationship was found between resilience of fundamental rights and freedom and neither resistance ($r(22)=0.21$, p=0.34) or recovery ($r(22)=0.16$, p=0.49). Furthermore, no relationship was found between resistance and recovery of fundamental rights and freedoms ($r(22)=0.31$, p=0.17). In other words, low fundamental rights and freedoms resistance corresponded to a wide range of recovery rates. Various countries reacted strongly at first, suggesting weak resistance, but were then quick to remove their restrictions, such as Estonia, which had the lowest resistance rate (the fastest escalation of strict measures), as well as a high recovery rate and the second highest overall fundamental rights and freedoms resilience. This was reflected by positively skewed data, evident for France, Norway, Estonia and the Czech Republic who adopted more stringent measures earlier on but were quick to remove them by the summer. In contrast, Sweden, which recorded the highest cumulative resilience value, showed a moderate resistance rate (9.10%) principally due to late measure introduction and a low recovery rate (−0.62%). Negatively skewed data were evident for Great Britain, Germany, Spain, Greece and Sweden who imposed and maintained more stringent measures later in the period. Other countries escalated quickly and maintained stringent government interventions during the whole period, indicating a low recovery, such as Portugal or Denmark. These mixed results reflect the different strategies adopted by countries to deal with the COVID-19 during the first wave.

### Economic system resilience
All countries experienced an economic decline, followed by recovery during the first wave, although to varying extents. These losses in turn translated in negative gross rate and a departure from the previous years. In this case, no country experienced similar values in the past 5 years suggesting that the robustness threshold was crossed for all 22 countries (online supplemental material 5 and 6), even though governments usually supported their economy. The cumulative resilience average for all 22 countries is −225.78. For the 20 countries that had both significant values for economic resilience and recovery (all except Slovenia and Slovakia), economic decline took on average 14 weeks to reach the economic trough. Economic resistance ($r(20)0=0.89$, p<0.001) and recovery ($r(20)=0.86$, p<0.001) were found to be associated with overall economic resilience: lower resistance and faster recovery resulted in higher cumulative GDP losses. Countries with the highest resistance such as Denmark (−0.30%), Norway (−0.40%), Finland (−0.44%) also had some of the highest overall economic resilience. Economic resistance and economic recovery were negatively correlated meaning that lower resistance was associated with higher recovery rate ($r(20)=0.84$, p<0.001).

### Multisystemic resilience
Cross-system comparisons of resilience results across health, fundamental rights and freedoms, and economic systems allowed us to identify potential linkages and trade-offs between systems and thereby derive a multisystemic understanding of resilience. Nine countries (Spain, UK, Italy, Belgium, Switzerland, Netherlands, Portugal, France and Sweden) had a peak above 40% during the first wave. Although several countries were robust at the health system level, none of the 22 European countries

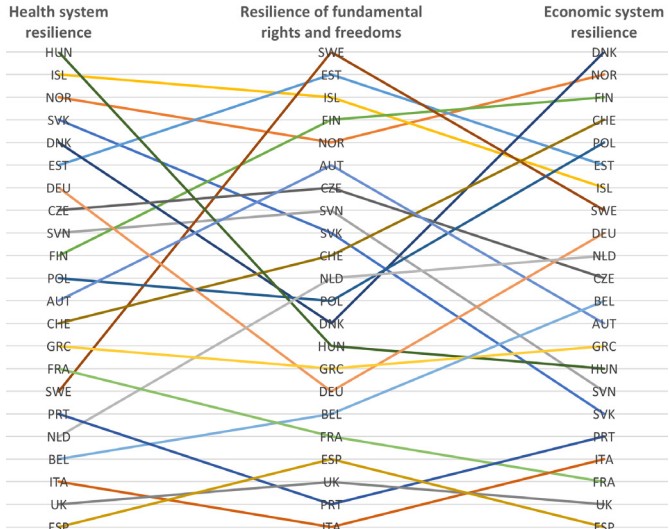

**Figure 2** Comparison of resilience ranking of health, fundamental rights and freedoms, and economic systems. AUT, Austria; BEL, Belgium; CZE, Czechia; DNK, Denmark; EST, Estonia; FIN, Finland; FRA, France; DEU, Germany; GRC, Greece; HUN, Hungary; ISL, Iceland; ITA, Italy; NLD, Netherlands; NOR, Norway; POL, Poland; PRT, Portugal; SVK, Slovak Republic; SVN, Slovenia; ESP, Spain; SWE, Sweden; CHE, Switzerland; UK, United Kingdom.

were robust across all systems, as all 22 countries crossed economic and fundamental rights and freedoms robustness thresholds. This aligns with the idea that when resilience is provided by intervention (and not by design), there is transfer of capacities from other systems.[22]

Health system resilience was associated with economic system resilience ($r(22)=0.69$, p<0.001); in other words, countries with higher excess mortality also suffered greater economic losses. Health system resilience appeared to be positively associated with overall fundamental rights and freedoms resilience but to a lesser extent ($r(22)=0.56$, p=0.007). This may be explained by the fact that many countries implemented stringent measures independently of their excess mortality trends. Finally, economic system resilience was found to be positively linked to fundamental rights and freedoms resilience ($r(22)=0.67$, p<0.001). In other words, the more restrictions on individual rights and freedoms, the more impact on the economy.

Some countries may exhibit a high level of resilience in one system but low resilience in another. When looking at the overall ranking based on countries' resilience in each system (figure 2), four countries were in the first quartile—Norway, Iceland, Estonia and Finland—and expressed high multisystemic resilience. By contrast, Spain, the UK, Italy, Portugal, France and Belgium were in the third quartile and expressed lower multisystemic resilience. We further established three main categories of country during the first wave of the COVID-19 pandemic. Figure 3 provides a graph for Norway, Germany and Spain to illustrate the three categories. It should be noted that these categories are not prescriptive nor rigid divisions, but rather constantly evolving trends as

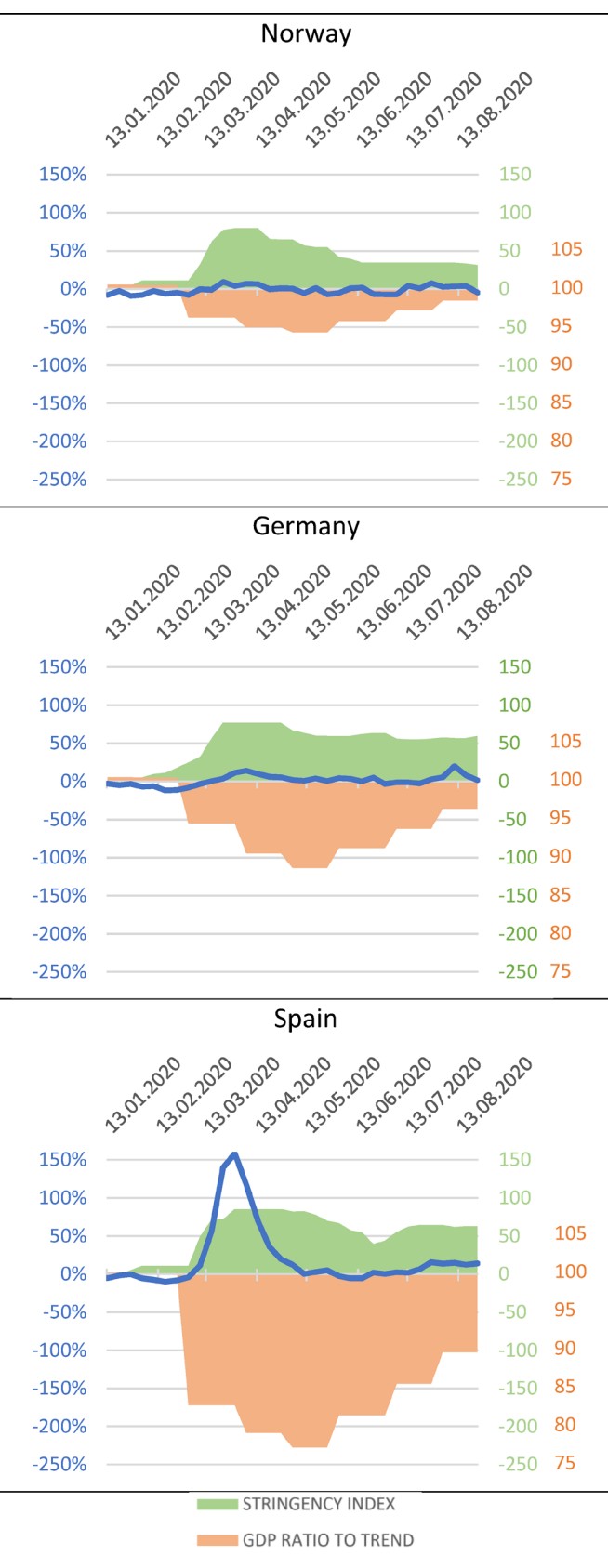

**Figure 3** Time series of selected countries representing the three categories and combining resilience measurements of the health, fundamental rights and freedoms, and economic systems.

resilience is an adaptive capacity. There are indications that countries changed category as the pandemic evolved.

A group of six countries (Hungary, Iceland, Norway, Slovakia, Denmark and Estonia) exhibited high health system resilience (except for Estonia, their excess mortality was negative). Estonia and Norway showed high level of resilience in all three systems, while Slovakia and Hungary had moderate resilience of fundamental rights and freedoms and economic resilience. Iceland adopted the second least stringent measures of the group yet also suffered moderate economic losses, potentially due to its greater dependence on the tourism industry.[23] By contrast, Denmark had a moderate resilience of fundamental rights and freedoms that was comparable with Hungary, yet better economic resilience.

A group of eight countries (Germany, Czechia, Slovenia, Finland, Poland, Austria, Switzerland and Greece) exhibited moderate health system resilience comprised between 43.68% and 119.68% of cumulative excess mortality. Except for Finland, which demonstrated high resilience of fundamental rights and freedoms, all of them exhibited moderate resilience in this system. There was more variability regarding economic resilience with high resilience in Switzerland, Poland and Slovenia and moderate resilience in the rest of the group. In other words, this group of countries usually suffered more severe economic downturns resulting from both stricter governmental measures and moderate health resilience. The heavier economic losses may also be related to greater dependence on tourism, as in Greece, Austria and Slovenia,[23] as well as greater levels of economic dependence on foreign trade and exports to other European countries hit by less demand from abroad, as with Germany, the largest European exporter.

Finally, eight countries (France, Sweden, Portugal, Netherlands, Belgium, Italy, UK and Spain) were part of a group with lower health system resilience with a cumulative excess mortality comprised between 251.33% and 653.18%. Five out of these eight countries were also countries with the lowest economic resilience. Except for Sweden, all countries exhibited either moderate (n=2) or low resilience (n=5) in fundamental rights and freedoms. Sweden's unique no-lockdown model resulted in the highest fundamental rights and freedoms resilience. By contrast, Italy, Portugal and the UK proved to be the less resilient in terms of fundamental rights and freedoms due in part to weaker crisis preparedness and slow reaction in the UK and potentially the balance of power towards the executive for Portugal and Italy.

## DISCUSSION
### Contribution to our understanding of resilience
This study operationalised the measurement of a multisystemic expression of resilience by assessing 22 European countries' resilience in three systems: health, the economy, and fundamental rights and freedoms. Our approach helps understand how countries are affected and respond to systemic crises by managing different societal objectives between systems.[24] The comparative analysis shows a significant tendency among all 22 European countries to favour health and economic systems over fundamental rights and freedoms during the first wave of the COVID-19 pandemic in Europe in 2020. Furthermore, while the recovery time has been used as the main resilience indicator,[4] our operationalisation of robustness, resistance and recovery in addition to an overall measure of resilience provide a more granular understanding of how countries were impacted and reacted to the first wave of the COVID-19 pandemic. For example, when considering only the recovery rate, the country with the most resilient health system would also be the one with the lowest resistance and the lowest overall health system resilience in our dataset (Spain).

Our results provide further evidence that the COVID-19 pandemic tested not only the resilience of health systems but also economic resilience and the resilience of fundamental rights and freedoms. The resilience capacities of one system may depend on and impact the others. It is not surprising that the implementation of stringent containment measures negatively impacted a country's economic performance. Nevertheless, several countries enforced relatively strict measures while limiting economic consequences. In contrast, the relationship between stringent government responses and health resilience is not as straightforward. Irrespective of their epidemiological situation, all governments adopted varying degrees of measures in early 2020, related to the novel character of the pandemic and the lack of public health capacities. By the end of August, no government could fully restore pre-crisis governance operating modes. Lenton *et al* found that prolonged stringent measures may not necessarily lead to improved health outcomes and may even reduce health system resilience in the long term due to decreased trust in government over time.[4] Finally, most countries in the dataset were more stringent than could be expected based on how liberal they usually are (online supplemental materials 7 and 8). Whether this contributed to a 'pendulum policy effect' that ultimately delayed responses during the second wave in autumn 2020 should be investigated.

### Limitations
The paper has some limitations regarding the conceptualisation of resilience. One of the main limitations of assessing resilience across countries is the issue of comparability; countries were not equally exposed to COVID-19 in early 2020 and did not start with the same context. Therefore, results showing high overall resilience do not necessarily imply the highest resilience capacity. In other words, our study measures in some ways how resilience capacities were expressed across different systems in relation to how they were exposed to the shock. Why and how it was expressed this way is another question that deserves more scrutiny. While this article seeks to capture a multisystemic understanding of resilience, our

analysis is limited to resilience in health, fundamental rights and freedoms, and economic systems. These systems were selected because they were more easily discerned and measurable, but systemic effects generated by the COVID-19 pandemic have also disrupted social, governance and environmental systems among others.[1] Furthermore, our approach explores only limited aspects of the multiple ways resilience can be expressed. The use of time series data highlights the capacity to cope and recover while overlooking the process of adaptation and transformation that is intrinsically important to the study of resilience.[1]

The proxy indicators used to measure health, fundamental rights and freedoms, and economic resilience have shortcomings. Regarding the GDP ratio-to-trend, the OECD stresses that it is more suitable for qualitative purposes to visualise the fluctuations of economic activity. Excess mortality was selected as it overcomes several issues related to the reporting of COVID-19 related deaths such as miscounting from misdiagnosis or under-reporting. However, it does not only include 'direct' COVID-19 mortality but also 'collateral damage' from other health conditions left untreated when the health system is overwhelmed. It also relies on the accuracy of mortality data from the 5-year period before the crisis, which is more or less accurate from country to country. The OxCGRT's stringency index to measure the resilience of fundamental rights and freedoms reflects the breaches in civil liberties that are intrinsic to the values of liberal democracies. However, the indicator does not reflect the respect of democratic processes nor the rule of law, essential components to assess broader democratic resilience. Limitations also arise as this indicator is a composite of additive unweighted indices that abstract away from nuances or any heterogeneity in the country responses. Its reliance on third party sources may also result in measurement bias.[25] Furthermore, unlike health and economic resilience, the stringency of measures does not simply reflect the 'country's inability to absorb and cope with the shock but also its ability to implement and modify measures quickly.

Additional limitations are related to measurements. First, resistance and recovery were approximated by using linear regression, but non-linear regression functions might provide a better fit to the data. Second, our analysis does not account for resilience overshoot (eg, when the measures adopted result in a highly negative excess mortality), nor the existence and length of a plateau between the resistance and recovery phases. Third, we did not compute a unique multisystemic measure of resilience encompassing all systems as this would have required a normative judgement on whether one system should be considered more important than another. This is fundamentally a political choice. While we recognise that directly comparing resilience in health, fundamental rights and freedom, and the economy can be challenging from a public health perspective, we believe that it is important to look at difficult questions of trade-offs. Such

assessment can help foster a democratic debate about the values in our societies and help select where investment needs to be made to increase our capacities to prevent such shocks.

Finally, some reflections are needed about the scope and applicability of the methodological approach. Given the fact that the COVID-19 pandemic has not been a single event but recurring waves of different magnitude, we believe that our approach should be applied to study resilience to the different waves of the pandemic. This will lead to an understanding of whether and how countries adapt their priorities and strategies over time. Ideally, assessing resilience should be based on several indicators in each system. Given the effectiveness of immunisation, one needs indicators that reflect the decoupling between mortality and infection. Another concern will be to expand the number of countries covered in the analysis. While the selected indicators are applicable to all countries, interpretations of the results should also consider the variety of cultural values and political systems in the world. Finally, there are also opportunities to use complementary methodological approaches such as interrupted times series to better relate what happened during the first wave of the COVID-19 pandemic and previous trends.[26]

## CONCLUSION

This study examined the multifaceted nature of resilience by investigating the inter-relationships between health, fundamental rights and freedoms, and economic systems. Our findings stress the consequences of limited pandemic preparedness, as no country demonstrated robustness in all three systems. Countries with lower capacity to rapidly control the pandemic compensated by implementing stringent measures, often at the expense of individual rights and freedoms. Notably, economic resilience was found to be closely intertwined with health outcomes and the stringency of measures, highlighting the interdependence and significance of resilience across systems. Furthermore, the recovery process exhibited substantial variations across the three systems and among different countries. This underscores the complexity and diversity of recovery efforts, suggesting that interventions aimed at enhancing resilience necessitate an understanding of trade-offs between systems with differing objectives. While further research is needed to deepen our understanding of multisystemic resilience, this study emphasises the importance of resilience as a focal point for improving governance capacities to effectively prevent, respond to and recover from current and future systemic shocks.[24]

**Author affiliations**
[1]Global Studies Institute, University of Geneva, Geneva, Switzerland
[2]Swiss Tropical and Public Health Institute, Basel, Switzerland
[3]Faculty of Law, University of Geneva, Geneva, Switzerland
[4]Department of Computer Science, Faculty of Science, University of Geneva, Geneva, Switzerland

**Contributors** DW, MC and AR conceptualised the multisystemic measurement approach to resilience. MC and AR analysed the data and developed the successive drafts of the paper. MC, AR and DW designed the figures, tables and supplementary material. All authors contributed content and comments to the paper. DW and NL are both principal investigators within the COVID-19 systemic crisis project. DW is responsible for the overall content as guarantor.

**Funding** This research was funded by the Swiss National Science Foundation (SNSF), within the scope of the COVID-19 systemic crisis project (Grant 31CA30_196396, https://data.snf.ch/COVID-19/snsf/196396).

**Competing interests** None declared.

**Patient and public involvement** Patients and/or the public were not involved in the design, or conduct, or reporting, or dissemination plans of this research.

**Patient consent for publication** Not applicable.

**Ethics approval** Not applicable.

**Provenance and peer review** Not commissioned; externally peer reviewed.

**Data availability statement** Data are available in a public, open access repository. All data used for this study are available in a public, open access repository. The sources of the data are the following: (1) health resilience: the Human Mortality Database (Reference 16), (2) resilience of fundamental rights and freedoms: Oxford COVID-19 Government Response Tracker, Blavatnik School of Government, University of Oxford (CC BY License) (Reference 17), (3) economic resilience: OECD Main Economic Indicators Publication (Reference 18).

**ORCID iDs**
Fabrizio Tediosi http://orcid.org/0000-0001-8671-9400
Chantal M Morel http://orcid.org/0000-0003-3984-9741
Didier Wernli http://orcid.org/0000-0002-1751-1961

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
