## [Reviewer comments · BMJ Open]

ARTICLE DETAILS

TITLE (PROVISIONAL)	Multisystemic resilience to shocks: a temporal analysis of health, fundamental rights and freedoms, and economic resilience during the first wave of the COVID-19 pandemic in 22 European countries
AUTHORS	Clausin, Mia; Rieckhoff, Alicia; Tediosi, Fabrizio; Morel, Chantal; Kaspiarovich, Yuliya; Levrat, Nicolas; Wernli, Didier

VERSION 1 – REVIEW

REVIEWER	Herman, Bumi Chulalongkorn University, Public Health
REVIEW RETURNED	09-Aug-2022

GENERAL COMMENTS	To the authors of the article It is a great article to review, and the authors justified the limitations accordingly and addressed other issues, including geographical advantage, the difference in time measurement and exposure, and the difference in countries' health and economic measures. However, here are some notes to be discussed 1. How applicable is the measurement of multisystemic resilience to shock to assess other countries' situations from the author's perspective? Even though the authors only focus on Europe, multisystemic resilience is linked to the global situation. Therefore, it is imperative to discuss this to boost the generalizability of the method.2. Since the study covers the first wave, how relevant is the measurement to assess the multisystemic resilience of the subsequent waves? The authors should address that the indicators of resilience, including health, may not be suitable if they only measure excess mortality. The procurement of vaccines and antivirus change the game. Hence excess mortality will no longer be relevant in the following years. Are any proposed indicators relevant to measuring multisystemic resilience in the future? For example, on page 11, the authors expressed concern about considering only recovery rate as a health resilience indicator. Will it be different if this rate is considered a health indicator for the current situation in 2022? I noticed that this concern is discussed on page 16 in brief. However, the authors may provide some hints for possible future studies.3. I prefer the interrupted time series analysis, which can accommodate specific measures or unprecedented events in the country that may affect the resilience indicators. The analysis could clearly depict how the countries tackled the first COVID
--

	wave. Although I accept the current analysis as it is not feasible to do the ITS analysis for 22 countries.
--	---

REVIEWER	Biundo, Eliana GSK Belgium
REVIEW RETURNED	12-Aug-2022

GENERAL COMMENTS	The original idea and objective of the paper is quite interesting, however i find the study to bring no contribution to the discussion on the topic and just providing conclusions not supported by data. In particular reflecting on resilience of human rights by assessing existence of non pharmaco interventions intended to save lives and contain a pandemic is not only inaccurate but also provides quite a negative message. Lastly, the indicators used to measure resilience for the three dimensions are a simplification but overall the choices to use them are well described. When the analysis goes into the 3 sub-categories of resilience I find that there is no additional contribution of this data except using an elaborate technique just to build credibility for an approach. Conclusions of the paper are not very informative and frankly based on a flawed methodology. I am sorry to say but without a serious reflection on the methodology and a more in depth reflection on the limitations of the chosen approach I do not think this manuscript should be published.
---

REVIEWER	Clark, Stephen University of Leeds
REVIEW RETURNED	27-Oct-2022

GENERAL COMMENTS	I was asked to conduct a statistical review of this article. Whilst it contains some simple linear regressions the contribution of statistical methods is light in the article. However here are my comments on reading the article. The article is well written and the quality of English used is excellent. Page 5 line 6 and 7. A GDP ratio below 100 does not necessarily signify contraction, just that the 'growth' (if any) is less than trend. Tables 2 and 3. Linear regressions are used to estimate the trend lines. Are the slopes reported significantly different from zero? Or isn't that of relevance? Page 6 line 29. unmatched ")" Page 7 line 12. Is this Great Britain or the United Kingdom. The latter includes Northern Ireland whilst the former doesn't. Page 7 line 18 and 19. Define which countries SVK and SVN are. Page 8 lines 28 and 30. What are the commas in the % in brackets? Should they be decimal points? Page 8 line 46. "excel" or "excess"? Page 9 line 40. unmatched ")"
--

REVIEWER	Vaillant, Michel Luxembourg Institute of Health, Competence Centre for Methodology and Statistics
REVIEW RETURNED	03-Nov-2022

GENERAL COMMENTS

The manuscript is an attempt to evaluate multisystemic resilience during the first wave of COVID-19 in some countries in Europe by defining indicators of resilience in health, fundamental rights and freedoms and economy. The authors smartly used previous theoretical publications to evaluate the burden of COVID-19 at the countries level.

The paper is well written. There are however some issues in the methods and the presentation of the results that need improvement. Methods should be extensively presented, particularly regarding the supplementary materials.

Major comments**Introduction:**

Page 2, line 55: it is mentioned resilience in health, economy and fundamental rights and freedoms. As Wernl (ref 14) is later on cited (page 3, line 5), the authors could explain why the governance and environmental components are not part of their study.

Methods:

Page 3, line 59: please define "GDP". Maybe worth adding a table of abbreviations.

Page 4, line 16-17: please sectors as title above "health system", "fundamental rights and freedoms" and "economic system" in table 1

Page 5: line 27: I was not able to find the cited definition of health robustness in the referenced publication

Page 5, line 28: please give a reference for the health robustness threshold. The reason for the choice of the value of 38% should be explained with solid arguments if no previous reference including validation of the cut-off above which define whether or not a country is robust. The level of excess mortality in Germany in 2018 is cited but why this country in particular and not another?

Page 5, line 31: please give a reference for the economic robustness threshold. The reason for the choice of the value of -1.25 should be explained.

Page 5, line 32: please give a reference for the stringency index robustness threshold. The reason for the choice of the value of 9.4 should be explained.

Page 5, line 42: for resistance linear regression was used to estimate the slope. It would be worth giving more explanation on the methodology. Specifically what are the outcome and the predictor, and what portion of data was used for the model. It is not clear whether the start date is the same for all countries.

	Page 5, line 56: same comment for recovery Page 6, line 8: health resistance, recovery and robustness could not be calculated for some countries. Some arbitrary value reflecting the reason for which the calculation was not possible should be looked for in order to enable comparisons with other countries. The fact is that these countries had a lower level of excess mortality and therefore it is important to consider them. Results: Page 6, line 18, 19, Table 3: the international organisation for standardisation country codes or any other classification may be used for the countries and indicated. Page 7, line 35-38: the sentence “For the 9 countries with a peak, both low level of resistance ($r(9) = .85, p < .001$), and high level of recovery ($r(9) = .78, p < .005$), were correlated with overall cumulative excess mortality and faster recovery (since only peaking countries showed high recovery)” is misleading as a positive correlation indicates that values of resistance and recovery increase with increasing values of cumulative excess mortality. Not only low levels of resistance or high level of recovery. Moreover resistance and recovery would be associated with faster recovery but it is not clear why what this faster recovery is. Minor comments Page 3, line 24: “selected” would be better than “chosen”
--	--

VERSION 1 – AUTHOR RESPONSE

Reviewer: 1

Dr. Bumi Herman, Chulalongkorn University, Hasanuddin University

R1C0 - To the authors of the article

It is a great article to review, and the authors justified the limitations accordingly and addressed other issues, including geographical advantage, the difference in time measurement and exposure, and the difference in countries' health and economic measures. However, here are some notes to be discussed

We thank the reviewer for the positive comments on the manuscript.

R1C1 - How applicable is the measurement of multisystemic resilience to shock to assess other countries' situations from the author's perspective? Even though the authors only focus on Europe, multisystemic resilience is linked to the global situation. Therefore, it is imperative to discuss this to boost the generalizability of the method.

We agree that there is a need to discuss multisystemic resilience from a global perspective. Given that chosen indicators are available and applicable to all countries, the methodological approach should be applicable to other countries and context. However, the interpretations of the results should be careful about the variety of cultural values and political system in the world. We added a new paragraph at the end of the discussion in which we discuss the applicability of the methodological approach. One sentence specifically covers the issue of expanding the scope to other countries.

R1C2 - Since the study covers the first wave, how relevant is the measurement to assess the multisystemic resilience of the subsequent waves? The authors should address that the indicators of resilience, including health, may not be suitable if they only measure excess mortality. The procurement of vaccines and antiviral change the game. Hence excess mortality will no longer be relevant in the following years. Are any proposed indicators relevant to measuring multisystemic resilience in the future? For example, on page 11, the authors expressed concern about considering only recovery rate as a health resilience indicator. Will it be different if this rate is considered a health indicator for the current situation in 2022? I noticed that this concern is discussed on page 16 in brief. However, the authors may provide some hints for possible future studies.

We believe that excess mortality is still a relevant indicator for other waves. Ideally, assessing resilience should be based on several indicators in each sector. Regarding health resilience, one need an indicator that reflects the decoupling between mortality and infection. There are several options here such as the number of cases or the hospitalization rate (they have both pros/cons). Furthermore, as the adoption of an effective technology is a recognized factor of enhanced resilience to shocks, immunization rate by an effective vaccines could be an indicator. We added two sentences that cover this aspect in the new paragraph about the applicability of the methodological approach in the discussion.

R1C3 - I prefer the interrupted time series analysis, which can accommodate specific measures or unprecedented events in the country that may affect the resilience indicators. The analysis could clearly depict how the countries tackled the first COVID wave. Although I accept the current analysis as it is not feasible to do the ITS analysis for 22 countries.

We are aware that interrupted time series analysis has been used to evaluate both the impact of the COVID-19 pandemic and interventions to tackle it. During the revision of the paper, we improved the measurement of robustness by comparing the event to previous trends. Hence, we totally agree with the reviewer that interrupted time series analysis would a very suitable approach for analysing the data as it allows us to relate what happened during the first wave of the COVID-19 pandemic in Europe and previous trends. Given the focus of the manuscript on several trends, it is not feasible here but might be included in another paper. We included a sentence in the discussion to say that improving our understanding of resilience might benefit from different approaches such as interrupted time series analysis.

Reviewer: 2

Dr. Eliana Biundo, GSK Belgium

R2C1 - Comments to the Author:

The original idea and objective of the paper is quite interesting, however i find the study to bring no contribution to the discussion on the topic and just providing conclusions not supported by data.

Here the main contribution of this paper is to show the relationship between resilience in different sectors in order to better understand societal resilience. Given the likelihood of future pandemics and other major shocks, there is an urgent need to design more resilient health systems capable of addressing a crisis while maintaining essential functions. The COVID-19 pandemic has shown that is costly and possibly not sustainable to rely on some approaches such as lockdowns. It mainly shows that countries, as unprepared as they were, had to rely to damaging strategies which may have a long-term impact on different sectors including health systems. Furthermore, the fact that countries with comparable level of shock exhibited different level of resilience in different systems created by the pandemic is an interesting finding. It creates an agenda for understanding what are the features of each country that make them able to cope with the shock the way they did. While there has been much attention to health system resilience, they have not been much discussion about considering the dynamics in different systems. We carefully read again our interpretation and conclusions and did not see anything that was not supported by the data.

R2C2 - In particular reflecting on resilience of human rights by assessing existence of non pharmaco interventions intended to save lives and contain a pandemic is not only inaccurate but also provides quite a negative message.

We believe it is time to recognize that the measures adopted to save life in the short-term also created some challenges for health systems and beyond. For example, Arsenault et. al¹ concluded that *“Given the widespread disruptions in health services demonstrated in this paper, many of which were unrelated to COVID-19 severity, our results call for rethinking pandemic preparedness and health system response. The unintended consequences of COVID- 19 responses may have outweighed the loss of life from COVID-19 itself, particularly in LMICs.”* Beyond health systems, which is the main interest here, there are also strong unintended consequences on other systems. While we live in a time of growing polarization of society and threat to democracy, the COVID-19 pandemic and associated response have put pressure on democracies worldwide. We believe these points are very important when it comes to invest money to make health system more resilient to shocks. While the reviewer points out the risk of negative message, we strongly believe that it is the role of science to look at difficult questions of trade-offs. Such assessment can help foster democratic debate about the values in our societies and were investment needs to be made to increase our capacities to prevent and react to such shocks.

To demonstrate further the relevance of our approach we conducted further analysis on the multisystemic resilience (the method are described in supplementary material 5). Based on how liberal countries usually are, the additional results shows that several countries were more stringent that they usually are during the first wave of the COVID-19 pandemic. These results are interesting because several countries also ended with lower-than-expected mortality after the first wave of the COVID-19 pandemic. As politics is often a pendulum, the perceived excess of stringency may have delayed timely action during the second wave of the COVID-19 pandemic in fall/winter 2020/2021 mostly due to the system delaying introduction of stringent measures in the early fall 2020. We strengthened the discussion and included the new results better to demonstrate the value of the approach taken.

R2C3 - Lastly, the indicators used to measure resilience for the three dimensions are a simplification but overall the choices to use them are well described. When the analysis goes into the 3 sub-categories of resilience I find that there is no additional contribution of this data except using an

¹ Arsenault C, Gage A, Kim MK, et al. COVID-19 and resilience of healthcare systems in ten countries. *Nature medicine* 2022 doi: 10.1038/s41591-022-01750-1

elaborate technique just to build credibility for an approach. Conclusions of the paper are not very informative and frankly based on a flawed methodology.

We agree that the indicators used are simplification but they reasonable approach given the constraint. Furthermore, there is currently a lot of interest in assessing and measuring resilience. Beyond measuring the resilience overall, what we show here is that resilience can be decomposed into several components which in turn provide relevant information about how a system cope with different phase of a shock. We believe this information is useful to understand that some countries were able to quickly escalate some measures or to understand the difference between the capacity to resist the shock and the capacity to recover from the shock. We believe that the results and discussion cover well this aspect. As written regarding the comment above, we do believe that our assumptions are timely and provide an adequate foundation for understanding societal resilience in its different aspects beyond one own's values.

R2C4 - I am sorry to say but without a serious reflection on the methodology and a more in depth reflection on the limitations of the chosen approach I do not think this manuscript should be published.

We understand the reviewer' concern and we try to consolidate our methodological approach (cf RC2). In addition, we described our assumptions/approach on which our methodological approach is based in a paper that was co-authored by 27 co-authors from different academic background, many of them have a background in public health.² Given the many unintended consequences of lockdown, we believe that we provide here a balanced approach about assessing resilience in different areas. It is then a matter of democratic debate and political choice to prioritise some areas over the others. That being said, we sought to improve several methodological aspects to address the comments and suggestions by the four reviewers. Given the more fundamental concern of reviewer 2, we strengthened the discussion section to cover the reviewer's concern. However, as some of the comments of the reviewer relates to core assumptions of about societal resilience, we recognize that not all comments were fully addressable.

Reviewer: 3

Dr. Stephen Clark, University of Leeds

R3C1 - Comments to the Author:

I was asked to conduct a statistical review of this article. Whilst it contains some simple linear regressions the contribution of statistical methods is light in the article. However here are my comments on reading the article. The article is well written and the quality of English used is excellent.

Thank you for the constructive comments on the manuscript.

R3C2 - Page 5 line 6 and 7. A GDP ratio below 100 does not necessarily signify contraction, just that the 'growth' (if any) is less than trend.

Thank you for spotting this issue. We corrected this in the manuscript.

R3C3 - Tables 2 and 3. Linear regressions are used to estimate the trend lines. Are the slopes reported significantly different from zero? Or isn't that of relevance?

² 1. Wernli D, Clausin M, Antulov-Fantulin N, et al. Building a multisystemic understanding of societal resilience to the COVID-19 pandemic. *BMJ Glob Health* 2021;6(7):e006794. doi: 10.1136/bmjgh-2021-006794

We added the statistical significance for the slope in what was table 3 (but now is supplementary material 6). This was very useful for analysis as we included only values that were statistically significant in the calculation of correlation between different measurements of resilience. We also consolidated our approach to compute the slope for resistance and recovery. This is described in more detail in the section of the methodological section about "measuring resilience, robustness, resistance and recovery).

R3C4 - Page 6 line 29. unmatched ")"

We corrected this.

R3C5 - Page 7 line 12. Is this Great Britain or the United Kingdom. The latter includes Northern Ireland whilst the former doesn't.

This was the United Kingdom. We corrected this in the manuscript.

R3C6 - Page 7 line 18 and 19. Define which countries SVK and SVN are.

We clarified this in the manuscript.

R4C7 - Page 8 lines 28 and 30. What are the commas in the % in brackets? Should they be decimal points?

We corrected this for decimal point.

R3C8 - Page 8 line 46. "excel" or "excess"?

We corrected the typo.

R3C9 - Page 9 line 40. unmatched ")".

We corrected this.

Reviewer: 4

Mr. Michel Vaillant, Luxembourg Institute of Health Comments to the Author:

R4C1 - The manuscript is an attempt to evaluate multisystemic resilience during the first wave of COVID-19 in some countries in Europe by defining indicators of resilience in health, fundamental rights and freedoms and economy. The authors smartly used previous theoretical publications to evaluate the burden of COVID-19 et the countries level.

We would like to thank reviewer#4 for his positive and constructive comments on the manuscript.

R4C2 - The paper is well written. There are however some issues in the methods and the presentation of the results that need improvement. Methods should be extensively presented, particularly regarding the supplementary materials.

We have strengthened the methodological section. In particular we revised how the resistance and recovery slopes are calculated. Several supplementary materials were added to the manuscript in relation to the measurement of robustness (further discussed below).

Major comments

R4C3 - Introduction:

Page 2, line 55: it is mentioned resilience in health, economy and fundamental rights and freedoms. As Wernl (ref 14) is later on cited (page 3, line 5), the authors could explain why the governance and environmental components are not part of their study.

We did not get data with enough time point for making the study comparable. We added an explanation about this in the introduction of the manuscript.

Methods:

R4C4 - Page 3, line 59: please define "GDP". Maybe worth adding a table of abbreviations.

We added a definition of GDP in the manuscript.

R4C5 Page 4, line 16-17: please sectors as title above "health system", "fundamental rights and freedoms" and , "economic system" in table 1

We implemented a more consistent use of how the sectors are named in the manuscript.

R4C6 - Page 5: line 27: I was not able to find the cited definition of health robustness in the referenced publication

The referenced publication discussed the concept of 'robustness'. We used the reference as an inspiration for defining robustness in the paper. The definition of robustness used in this paper is presented in table 1 and is different from the reference. Robustness implies that a system is not able to maintain its usual performance. We clarified this in table 1.

R4C7 - Page 5, line 28: please give a reference for the health robustness threshold. The reason for the choice of the value of 38% should be explained with solid arguments if no previous reference including validation of the cut-off above which define whether or not a country is robust. The level of excess mortality in Germany in 2018 is cited but why this country in particular and not another?

As mentioned above, robustness implies that a disruption is affecting the system in such a way that normal time performance is reduced. In the case of health system, this translates into a reduction of the provision of health services³ and ultimately in variation of some health outcomes indicators such as excess mortality. While there is growing number of studies on health resilience⁴, there is not a specific threshold defined in the literature. Defining robustness should logically refers to what are normal performance, but we agree that the 38% was not enough grounded in the literature.

To address the reviewer's comment, we revised our methodological approach to better identify whether the level of excess mortality measured in different countries during the first wave of the COVID-19 pandemic in Europe were outliers considering the values from 2015-2019. We used the interquartile range (1.5*IQR) to calculate whether the value from 2020 were outliers. In addition, we also calculated Z-score.

Basing the assessment of robustness on statistical measurements may have some limitations. It implies that countries are robust as long as they are not outliers. This seems acceptable here as there was no or limited report of reduction of health services before the pandemic (2015-2019). However, it should be noted the absence of reporting does not mean that there was no disruption at all.

³ Arsenault C, Gage A, Kim MK, et al. COVID-19 and resilience of healthcare systems in ten countries. *Nature medicine* 2022 doi: 10.1038/s41591-022-01750-1.

⁴ Fleming P, O'Donoghue C, Almirall-Sanchez A, et al. Metrics and indicators used to assess health system resilience in response to shocks to health systems in high income countries—A systematic review. *Health Policy* 2022;126(12):1195-205. doi: <https://doi.org/10.1016/j.healthpol.2022.10.001>

Furthermore, the assumption that identifying outliers is a good measure of robustness is supported by emerging evidence of strong disruptions of access to health services during the COVID-19 pandemic. The most interesting methodological approach is the one by Arsenault et al (cf. ref 3) who have quantified disruptions in 10 countries but none of these countries is present in the dataset. The new methodological approach has led to a revision of the methodological and results sections. In addition, we added supplementary material 1 and 2.

R4C8 - Page 5, line 31: please give a reference for the economic robustness threshold. The reason for the choice of the value of -1.25 should be explained.

We adopted the same approach than for health resilience (see comment C4R7) using both IQR (and Z-score) to determine whether countries were outliers. Based on data on the annual growth rate by the World Bank⁵, we clearly show that all countries in the dataset experienced disruptions that were unusual compared to the previous years. In that sense, we infer that countries were not robust on the economic level. We adapted the methodological and results section to describe the changes described above and added supplementary material 3 and 4.

R4C9 - Page 5, line 32: please give a reference for the stringency index robustness threshold. The reason for the choice of the value of 9.4 should be explained.

Regarding the measure of fundamental rights and freedoms, the situation is different as the Oxford stringency index was introduced with the pandemic and there is not point of comparison with the period preceding the pandemic. In this case, we first looked for indicators that may reflect such as liberal democracy index or an indicator on restrictions to liberty of freedom of assembly. In the first case the liberal democracy index is much broader than the Oxford Stringency Index, but it tells us that except for Hungary and Poland which where outliers, the 20 other countries had a comparable baseline. In the second case, the indicator was too specific and show little variation while measures restricting the freedom of assembly were implemented according to the stringency index. Given the limited relevant data about previous, the approach used for economic and health robustness cannot be replicated.

However, given that the Stringency Index has been computed from January 2020, when little measures existed in the first place, they provide a baseline for comparison, which even more relevant given the comparable value for the state of democracy in Europe except for Hungary and Poland (as measured by the liberal democracy index from V-dem). Taking the Oxford Stringency index, it is evident that countries implemented widespread restrictions and were not robust in the area of fundamental rights and freedoms. Even the lowest maximum score of 46,3 which was from Sweden implies several restrictions on fundamental rights and freedoms such as no public gatherings of more than 50 people etc. To further evaluate how countries compared to each other, we calculated the cumulative stringency for the week 3 to week 35 and then the average for all countries. Given the variability in the dataset, Sweden did not appear as an outlier based on $1.5 \times \text{IQR}$ but it has Z-score of 2,06.

We included the new method, results and discussion in the manuscript.

R4C10 - Page 5, line 42: for resistance linear regression was used to estimate the slope. It would be worth giving more explanation on the methodology. Specifically what are the outcome and the predictor, and what portion of data was used for the model. It is not clear whether the start date is the same for all countries.

⁵ <https://databank.worldbank.org/reports.aspx?source=2&series=NY.GDP.MKTP.KD.ZG&country=>

We reworked the paragraph to better explain what we did. As data does not perfectly align, the linear regression allowed us to consider the average increase (regarding resistance) or decrease per week. We clarified that the start date was not the same for each country and reworked the table as per comments R3C3.

R4C11 - Page 5, line 56: same comment for recovery

We did the same approach than for R4C10. In other words, we did not use the same date for all countries but from the peak until the lowest value during week 3 to 35 in 2020. This has been clarified in the manuscript.

R4C12 - Page 6, line 8: health resistance, recovery and robustness could not be calculated for some countries. Some arbitrary value reflecting the reason for which the calculation was not possible should be looked for in order to enable comparisons with other countries. The fact is that these countries had a lower level of excess mortality and therefore it is important to consider them.

We recognize that it will be useful. The concept of robustness, resistance and recovery only applies when there is visible disruption. In the originally submitted version of the manuscript, countries were excluded because they did not have excess mortality, so measuring a slope was difficult.

This is where the calculation of the area behind the curve becomes a good substitute because it encompassed the many fluctuations.

Given the comments by other reviewers and the introduction of simple rules for the start and endpoint of the slope, we were able to calculate slope for all countries (CF table 3) and derived their statistical significance. If the value for the slope were not significant (taking >0.05 as a threshold), we did not include those values for calculating association between different measurements.

Results:

R4C13 - Page 6, line 18, 19, Table 3: the international organisation for standardisation country codes or any other classification may be used for the countries and indicated.

We used the World Bank Classification and improved the consistency across the manuscript.

R4C14 Page 7, line 35-38: the sentence "For the 9 countries with a peak, both low level of resistance ($r(9) = .85, p < .001$), and high level of recovery ($r(9) = .78, p < .005$), were correlated with overall cumulative excess mortality and faster recovery (since only peaking countries showed high recovery)" is misleading as a positive correlation indicates that values of resistance and recovery increase with increasing values of cumulative excess mortality. Not only low levels of resistance or high level of recovery. Moreover resistance and recovery would be associated with faster recovery but it is not clear why what this faster recovery is.

This is an important point and further looking at the results, we rephrased the sentence in the manuscript.

Minor comments

R4C15 - Page 3, line 24: "selected" would be better than "chosen"
We replaced 'chosen' by 'selected'.

VERSION 2 – REVIEW

REVIEWER	Herman, Bumi Chulalongkorn University, Public Health
REVIEW RETURNED	18-Mar-2023

GENERAL COMMENTS	The authors addressed our concerns accordingly and stated further actions that can be done to increase the robustness of the methodology
--

REVIEWER	Vaillant, Michel Luxembourg Institute of Health, Competence Centre for Methodology and Statistics
REVIEW RETURNED	30-Mar-2023

GENERAL COMMENTS	I am comfortable with the manuscript as it is now. The authors have taken the comments from the reviewers into account and the manuscript has greatly improved.
---

VERSION 2 – AUTHOR RESPONSE

Multisystemic resilience to shocks: a temporal analysis of health, fundamental rights and freedoms, and economic resilience during the first wave of the COVID-19 pandemic in 22 European countries.

C0 - Thank you very much for submitting your revised manuscript to BMJ Open. Unfortunately, only two of the previous reviewers agreed to review the revision, so your revised manuscript and rebuttal was reviewed in-house by the editorial team and by an Associate Editor.

We would like to thank the editorial team and the Associate Editor for reviewing the manuscript and providing further useful and constructive comments.

C1 - Whilst we appreciate that the remaining comments from the reviewers (below) are positive, we would still need you to satisfactorily address the following points:

- The title doesn't clearly state what the research question is and doesn't describe the study design or what the settings are (it should probably clarify that you are looking at 22 European countries)

We slightly adapted the title based on the suggestion. Regarding the study design, we looked at paper doing similar things regarding the temporal evolution of excess mortality or other relevant health indicators⁶. Based on this paper, we suggested the following title: "Multisystemic resilience to

⁶ For example, <https://www.ncbi.nlm.nih.gov/pmc/articles/PMC7453680/>, and <https://smw.ch/index.php/smw/article/view/3111/5188>

shocks: a temporal analysis of health, fundamental rights and freedoms, and economic resilience during the first wave of the COVID-19 pandemic in 22 European countries”.

C2 - There also isn't a very clear research question in the abstract (it appears to be more of a 'methodology' paper which makes the article's scope as a research article questionable).

We reformulated several parts of the abstract to make the scope more compatible with a research paper.

C3 - Can you reformat the abstract so that it is using the (relevant) headings suggested in the journal's instructions for authors for research articles? See: <https://bmjopen.bmj.com/pages/authors#research>

We reorganized the abstract based on the relevant sections from <https://bmjopen.bmj.com/pages/authors#research> and comment C2 above.

C4 - The abstract doesn't need to include the Patient and Public Involvement (PPI) statement, so please remove this. The PPI statement just needs to be included in the methods section. You also say: “This study is based on an online search of publicly available data”, which is vague. What data was used for this study?

We removed the statement from the abstract and slightly reformulated the statement to clarify what data are used.

C5 - Can you please work on making the abstract >> results and conclusions sections more informative? Currently, the conclusion section is focused on the novelty of the methods used (please note, novelty is not a publication criterion for BMJ Open); can you clarify what the actual findings are and why these findings are useful?

We reworked the results and conclusion sections of the abstract. We clarified what is the conclusion of this study (none was country was resilient in the three dimensions of resilience) and included a sentence on how this can be useful to prepare for the next pandemic.

C6 - In general, please can you work again on making this article more accessible and readable for BMJ Open's general readership? We found the paper very difficult to follow in places (we felt your related paper was a lot more readable: Building a multisystemic understanding of societal resilience to the COVID-19 pandemic <https://gh.bmj.com/content/6/7/e006794>).

We reread the paper and try to simplify and improve the writing of several sections. For example the conclusion section was rewritten. In addition, we streamlined the use of system in the manuscript (and removed the words 'sector', 'domain' and 'dimensions') to clarified that we are referring to the resilience of systems. Overall, we reduced the word length in the main text from 6769 in the resubmitted version of the manuscript to 5004 (equivalent to 26% reduction).

C7 1. Better organizing the methods section and using the STROBE guidelines to report the methods section (we didn't see a reporting guideline).

Based on the relevant sections of the STROBE guidelines, we reorganized the methods sections as follows 1) study design and setting, 2) data sources, 3) measurements and 4) statistical methods (the participants section was not applicable).

C8 - The methods section wasn't very clear about what datasets were used and how you searched for publicly available datasets. You clarify what variables were selected but it wasn't clear to us how the data were obtained and whether data extraction was done systematically. Why were these 22 countries selected, specifically? What are the inclusion criteria?

In section 2.2 about study design, we added 1) a sentence to explain how we systematically extracted the data for each country in a spreadsheet and 2) clarified the inclusion criteria at the end of the paragraph.

C9 - We felt the presentation of the methods was poor. For example, a description of the statistical analysis is provided in a section with the heading 'Measuring resilience, robustness, resistance, and recovery', Can you use more sub-headings to guide the reader? You say you carried out a time series analysis but we missed what this actually involved. Reviewer 3 previously noted you just carried out some simple linear regressions.

We added a sub-heading statistical analysis. Regarding time series, the definition of time series is "*a series of values of a quantity obtained at successive times, often with equal intervals between them.*" [Oxford Dictionary of English]. At different places in the manuscript, we clarified that we are using time series data but are not conducting time series analysis such as Autoregressive (AR), Moving Average (MA), Autoregressive Integrated Moving Average (ARIMA), Vector Autoregression (VAR), and Hierarchical time series models).

While rereading the manuscript, we stumbled upon an inconsistency in the way we calculated resilience as the area under the curve. To avoid inconsistencies, we clarified that we are using the cumulative values for the selected metrics (excess mortality, stringency index, and GDP ratio to trend. We recalculated the Pearson coefficient with no major changes in the results. We subsequently simplified the methodology section to reflect these changes. The figures were also adapted to reflect the latest calculations.

In addition, we clarified the method used for outlier detection and corrected an error regarding the number of countries that crossed the health system robustness threshold. We also corrected an error in the table about robustness in health system (only 6 countries were robust based on 1.5 interquartile range method). While checking the calculations and results again, we discovered and corrected two errors with no impact on the interpretation of the results. First is regarding the Person coefficient between health resistance and health recovery (for the 10 countries with significant values for both measurements). The second is regarding the values used for deriving the Z-score for the stringency index that were not aligned with those used in the manuscript. Finally, we reorganised and reworked some of the description of the supplementary material to make clearer the method used.

C10 - 2. Shortening and summarizing the results section, which is too long and refer readers to tables and figures as much as possible.

We summarized and reduced the length of the results section by removing the listing of countries and adding tables where the readers can see the results from themselves for the sections on health, economic and fundamental rights and freedoms resilience. We believe this makes the result section simpler and clearer for the reader. The section on multisystemic resilience was rewritten to integrate the section of the discussion on the typology of countries. In addition, several parts of the result sections were cut.

C11 - 3. Making some cuts in the discussion section, which we also felt was too long. Again, it would be helpful to provide side-headings to guide the reader. Our LforA suggests the following: "We also recommend, but do not insist, that the discussion section is no longer than five paragraphs and follows this overall structure (you do not need to use these as subheadings): a statement of the principal findings; strengths and weaknesses of the study; strengths and weaknesses in relation to other studies, discussing important differences in results; the meaning of the study: possible explanations and implications for clinicians and policymakers; and unanswered questions and future research." (<https://bmjopen.bmj.com/pages/authors#research>)

We made some substantial cuts to the discussion (for example integrating the section on typologies of resilience in the result section) and added sub-headings to clarify the structure. While we did not reorganize the discussion to follow exactly the template that has been provided, we believe that the discussion better covers the main point mentioned in the comment above.

C12 - Please also carefully proofread the paper one more time. There are various typos that need correcting e.g. page 8: " Given the level the unprecedented level of restrictions implemented in all countries,.."

Thank you again for the useful and constructive comments, we proofread the paper and corrected several English issues.